# Beverage Consumption and Factors Influencing the Choice of Beverages among Polish Children Aged 11–13 Years in 2018–2023

**DOI:** 10.3390/nu16111625

**Published:** 2024-05-26

**Authors:** Julianna Kostecka, Izabella Jackowska, Izabela Chabros, Joanna Kostecka-Jarecka, Paulina Kawecka, Malgorzata Kostecka

**Affiliations:** 1Faculty of Medicine, Medical University of Lublin, Chodźki 19, 20-093 Lublin, Poland; kostecka.julianna@gmail.com; 2Department of Chemistry, Faculty of Food Science and Biotechnology, University of Life Sciences, Akademicka 15, 20-950 Lublin, Poland; izabella.jackowska@up.lublin.pl (I.J.); paulina.kawecka@up.lublin.pl (P.K.); 3Student Scientific Society of Dietitians, Faculty of Food Science and Biotechnology, University of Life Sciences, Akademicka 15, 20-950 Lublin, Poland; 4Department of Paediatrics, Lung Diseases and Rheumatology, University Children’s Hospital in Lublin, Antoniego Gębali 6, 20-093 Lublin, Poland

**Keywords:** hydration, beverage consumption, taste preferences, sugar-sweetened beverages

## Abstract

Adequate hydration is essential for good health, and an individual’s hydration status is determined by the quantity and type of ingested fluids. The aim of the present study was to determine the hydration status of school-age children and evaluate changes in the type and quantity of consumed beverages between 2018 and 2023. The study was conducted in two stages between 2018 and 2023, and a total of 1030 fully completed questionnaires were returned by the children and their parents. A comparison of the parents’ responses regarding factors that affect beverage choices revealed that beverage composition was more significant for the parents in 2023 than in 2018, whereas health-promoting properties were significant for only less than 30% of the respondents. Taste preferences were important for both the parents and the children, and they were the main criterion in the choice of beverages in both 2018 and 2023. In turn, advertising was an important factor for children, and the percentage of children who were guided by advertising in their choice of beverages increased from 52.1% in 2018 to 58.5% in 2023 (*p* < 0.05). Daily fluid intake from beverages in children aged 11–13 years generally does not meet recommended intakes. Low fluid intake can negatively affect children’s hydration status and bodily functions. Taste preferences and advertising were correlated with a higher intake of carbonated and non-carbonated sugar-sweetened beverages (SSBs) and dairy beverages. The percentage of children who bought drinks independently and had access to SSBs increased significantly during the analyzed period. Obtain results indicate that nutrition education programs are needed to teach adolescents to make healthy drink choices, limit their consumption of SSBs and EDs, and promote regular intake of natural mineral water and non-sweetened dairy beverages.

## 1. Introduction

Adequate hydration is essential for good health, and an individual’s hydration status is determined by the quantity and type of ingested fluids. The demand for water and fluids is influenced by age, sex, physiological status, and physical activity. The sensation of thirst is a warning sign that the body is deficient in water, and dehydration is particularly dangerous for children. For children weighing 11 kg to 20 kg, the daily water requirement is 100 mL/kg for the first 10 kg and 50 mL/kg for every kg above 10 kg. For children weighing more than 20 kg, fluid requirement is calculated as 1500 mL for 20 kg and 20 mL/kg for every kg above 20 kg, but more than 2400 mL of fluid should not be administered at once [1]. Water can be taken in the form of various liquids and also with food (in Poland up to 0.7 L/day on average). In the diet of an average adult Pole, the primary source of water is beverages, including water, juices or tea, which account for more than 70% of the daily intake from the total pool of fluids. This is followed by vegetables and fruit, which provide about 18%, and milk and dairy drinks covering about 10% of the daily intake. Water (bottled water or tap water) is the best choice to meet the fluid requirements of a group of children. The consumption of bottled water in Poland continues to increase, and it was determined at 114 L per capita in 2019, approaching the European average. Despite this rapid increase, the consumption of bottled water in Poland is still nearly half lower than in Italy or France [2,3]. Due to a wide selection of beverages on the market, consumers, in particular children’s parents and guardians as well as teachers and school staff, should be provided with specific recommendations regarding different beverages’ potential to affect hydration status, supply energy and other nutrients, as well as their negative effects on the body [4,5]. Children’s diets are increasingly abundant in sugar, which can be partly attributed to higher consumption of sugar-sweetened beverages (SSBs) [6,7]. The intake of SSBs is correlated with a child’s age because children receive money from their parents to buy soft drinks and snacks in shops and vending machines that are widespread in Polish schools. It has been well documented that SSBs have harmful effects on children’s health by contributing to obesity, dental caries or diabetes [8], and their excessive consumption may be associated with emotional and behavioral problems, in particular, attention deficit hyperactivity disorder (ADHD) [9]. Higher intake of SSBs also decreases children’s consumption of milk and dairy products, which gives particular cause for concern because milk contains calcium that promotes bone mineralization [10,11]. Milk is considered beneficial for children’s growth and development because it is an excellent source of essential nutrients [10]. Different studies have shown that children’s total caloric intake can be decreased by replacing SSBs with water [12,13]. Therefore, children should be encouraged to drink water instead of high-calorie drinks [14,15]. Fruit juice offers a healthier alternative to SSBs because it contains health-promoting vitamins and minerals. In the context of a healthy dietary pattern, there is evidence to suggest that only 100% fruit juice can provide beneficial nutrients without contributing to obesity in children [16]. It is very important to promote healthy dietary choices in children relating to the consumption of different types of liquids and enhance the preference for water and unsweetened beverages.

The aim of the present study was to determine the hydration status of school-age children and to evaluate changes in the type and quantity of consumed beverages between 2018 and 2023. The questionnaire survey provided information about the factors that affect the choice of beverages among parents and children and children’s intake of different beverages, including pure water (bottled water or tap water). Children’s habits and acquired consumption patterns were analyzed, and special attention was paid to whether children have access to wa-ter and beverages in school, whether Another important research objective was to assess whether children meet their fluid requirements and whether physical activity and body weight, expressed as the body mass index (BMI), influence the amount and type of fluids consumed by children. Children’s habits and acquired consumption patterns were analyzed, and special attention was paid to whether children have access to water and beverages in school, whether parents provide physically active children with an adequate supply of fluids, and whether children ask for a drink when they are thirsty. Children’s beverage preferences, as well as changes in children’s preferences and consumption of various beverages, were examined in the analyzed period.

## 2. Materials and Methods

### 2.1. Study Design

The present study was designed as a cross-sectional study, and it was conducted in two stages between 2018 and 2023. Before the study, the research participants were provided with all the relevant information, which was given to students during lessons at school and to their parents or legal guardians during a meeting. All data were collected by experienced researchers (well-trained in collecting dietary data) at schools during meetings that replaced regular school lessons. Students who refused to participate in the study attended other school activities. Teachers were present during the research, but they did not take part in data collection. All parents signed informed consent forms concerning their and their children’s participation in the research. The first part of the study was conducted between September and December 2018, and it involved 491 children aged 11–13 years who attended primary schools in the voivodeships of Lublin (*n* = 288, 58.7%), Mazovia (*n* = 102, 20.8%), and Podkarpacie (*n* = 101, 20.5%). Boys accounted for 46%, and girls accounted for 54% of the study group. The questionnaires were completed by 454 mothers and only 37 fathers. The second part of the study was conducted between October 2022 and March 2023, and it involved 539 children aged 11–13 years who attended primary schools in the voivodeships of Lublin (*n* = 267, 49.5%), Mazovia (*n* = 134, 24.8%), and Podkarpacie (*n* = 138, 25.7%). The study group consisted of 45% boys and 55% girls. All questionnaires were completed by mothers; therefore, the parent’s sex was disregarded in the discussion of the results. 

### 2.2. Characteristics of the Study Population

Children aged 11–13 years and their parents were invited to participate in the research (Table 1). Children were eligible to take part in the study if they were aged 11–13 years, resided in one of the three voivodeships in central–eastern Poland (Lublin, Podkarpacie, and Mazovia), and had no history of disease requiring dietary modifications (allergies, type 1 diabetes, kidney diseases, or Crohn’s disease).

### 2.3. Questionnaire for Data Collection

The children and the parents filled in identical questionnaires composed of 25 questions. The questionnaires differed only in terms of the personal data section. Children were asked to indicate their age, sex, and physical activity levels, whereas the parents were asked to indicate their age, sex, place of residence (urban or rural area), education (university, secondary, vocational), and the child’s body mass and height. The questionnaire was designed by the authors, and it contained single-choice and multiple-choice questions concerning the types of beverages consumed by children (natural mineral water include tap water, carbonated mineral water, flavored mineral water, non-carbonated SSBs, carbonated SSBs, energy drinks, juice, tea and other sweetened hot beverages, milk and dairy beverages), factors that affect parents and children’s choice of beverages, daily fluid intake, fluid intake in school, physical activity, sedentary behaviors, serving size in household measures, size and type of liquid packaging, types of fluids consumed during meals and at school, types of beverages purchased independently by children, parental control over the type and quantity of beverages consumed by children, experiences of thirst, and the preferred types of beverages. The participants rated their responses on a 5-point Likert scale (5—very often, 4—often, 3—moderately, 2—rarely, 1—very rarely). Children’s weight and height data were used to assess their nutritional status with the use of a growth percentile calculator [17].

The questionnaire survey was administered during school hours (mostly homeroom time), and children completed paper questionnaires independently under the researcher’s supervision. The students were informed about the purpose of the questionnaire and were provided with instructions before the survey. The parents completed the questionnaires independently at home, and the filled-out questionnaires were returned by the children to their homeroom teachers. On average, the students completed the questionnaires within 24 min, and the parents returned the filled-out questionnaires within 10 days. A total of 1030 fully completed questionnaires were returned by the children and their parents.

### 2.4. Data Analysis

The results were presented in tables and figures and were described. Categorical variables were presented as sample percentages (%). The differences between groups were analyzed in the chi-squared test (categorical variables). Before statistical analysis, data were checked for normal distribution in the Kolmogorov–Smirnov test. The odds ratios (ORs) and 95% confidence intervals (95% CIs) were calculated. The reference categories (OR = 1.00) included adherence to the consumption of natural mineral water and adherence to selected demographic factors (sex, body mass) and parental factors (education, age, and place of residence). The ORs were adjusted for factors influencing beverage choice and types of beverages. The significance of ORs was assessed by Wald’s statistics. The results of all tests were regarded as statistically significant at *p* < 0.05. Data were processed in the Statistica program (version 13.1 PL; StatSoft Inc., Tulsa, OK, USA; StatSoft, Krakow, Poland).

## 3. Results

In 2018, average fluid consumption in the study group was determined at 1606 mL/day (min 1200 mL/day–max 2710 mL/day), including natural and carbonated mineral water, SSBs, tap water, juice, tea and other hot beverages, milk and dairy beverages, energy drinks (EDs), and diet (light) beverages. On average, boys consumed 210 mL more fluids than girls (1794 mL/day vs. 1583 mL/day; *p* < 0.05). The average fluid intake in both sexes was below the recommended levels for this age group (2100 mL/day for boys and 1900 mL/day for girls). In 2023, the average fluid intake in the analyzed population increased by 286 mL and reached 1890 mL/day (min 1050 mL/day–max 2950 mL/day). On average, boys consumed 250 mL more fluids than girls (1961 mL/day vs. 1711 mL/day; *p* < 0.05). In summary, only 25% of the children met daily fluid intake requirements from drinks consumed in 2018. In 2023, only 21% of the respondents met these requirements despite an increase in daily fluid intake.

A comparison of the parents’ responses regarding factors that affect beverage choices (Table 2) revealed that beverage composition was more significant for the parents in 2023 than in 2018 (*p* < 0.05), whereas health-promoting properties were significant for only less than 30% of the respondents. Taste preferences were important for both the parents and the children, and they were the main criterion in the choice of beverages in both 2018 and 2023. In turn, advertising was an important factor for children, and the percentage of children who were guided by advertising in their choice of beverages increased from 52.1% in 2018 to 58.5% in 2023 (*p* < 0.05). Between 2018 and 2023, a worrying increase was noted in the intake of EDs and diet (light) drinks, especially among children, and children demonstrated a greater preference for SSBs. 

In the studied period, the number of children consuming natural mineral water and juice decreased by 11.5% and 23.3%, respectively (*p* < 0.05), whereas a significant increase was noted in the number of children consuming carbonated SSBs (41.4%, *p* < 0.05) and EDs (32%, *p* < 0.05) (Table 2, Figure 1).

The size of beverage packaging was also considered in the analysis. In 2018, the average packaging size was 346 mL ± 126 mL, and beverages sold in 333 mL (flavored mineral water, juice) and 500 mL bottles (natural mineral water, carbonated SSBs) were most often selected. In 2023, the average packaging size increased to 475 mL ± 131 mL, and beverages sold in 500 mL bottles (natural mineral water, non-carbonated SSBs) and 1000 mL bottles and cartons (juice, carbonated SSBs) were the most popular. 

The intake of natural mineral water was highest during physical activity (physical education classes and extracurricular activities). On average, children participating in extracurricular sports consumed 550 mL ± 119 mL of water in 2018 and 640 mL ± 130 mL of water in 2023 (*p* < 0.05). Water intake was significantly higher among physically active boys than in girls who took part in additional physical activity in both 2018 (655 mL ± 75 mL vs. 420 mL ± 105 mL; *p* < 0.001) and 2023 (750 mL ± 96 mL vs. 510 mL ± 115 mL; *p* < 0.001). Children with a sedentary lifestyle were more likely to reach for sugar-sweetened soft drinks, and the consumption of these beverages increased significantly in the analyzed period. Carbonated SSBs, juice, and mineral water were most frequently consumed during school trips and class events in 2023. Girls had a significantly higher preference for juice than boys (1.42; 95% CI 1.11–1.59, *p* < 0.01), whereas boys were significantly more likely to reach for carbonated SSBs (1.35; 95% CI 1.09–1.56, *p* < 0.05) and EDs (1.31; 95% CI 1.07–1.49, *p* < 0.05) than girls. Sugar intake was also affected by the amount of sugar added to hot beverages. In 2018, 50% of girls and around 40% of boys added one teaspoon of sugar to hot drinks, and a third of the children added two teaspoons of sugar (sex was not a differentiating factor, *p* > 0.05). Boys were less likely to add sugar to hot drinks than girls. In 2023, 40% of children of both sexes added one teaspoon of sugar to hot drinks, but fewer boys than girls added sugar to hot drinks relative to 2018.

To prevent dehydration, water should be consumed regularly before the sensation of thirst appears. A total of 45% of the parents agreed with the above statement, and no significant differences in the percentage of answers were observed between 2018 and 2023 (*p* > 0.05). More than a third of the surveyed girls and 25% of the boys did not report feelings of thirst to their parents and did not request a drink when they were thirsty. Approximately 22% and 24% of the children consumed beverages at night-time in 2018 and 2023, respectively, and girls consumed beverages at night significantly more often than boys (*p* < 0.05). Tea, juice, natural mineral water, tap water, and carbonated SSBs were most frequently consumed at night. Most children did not consume beverages regularly, and no significant differences in the percentage of answers were observed between the years. In 2018, boys were more likely to drink spontaneously, in particular, during physical activity (*p* < 0.05), whereas in 2023, spontaneous fluid intake was similar in both sexes, but it occurred more frequently at home and during extracurricular activities than in school. 

A comparative analysis revealed that taste preferences, advertising, and the awareness that hydration is essential for good health exerted a greater influence on the intake and consumption frequency of natural mineral water than SSBs and EDs (Table 3). The awareness that water delivers health benefits decreased the consumption of SSBs in 2023 relative to 2018 (*p* < 0.05). Factors such as sugar content, price, and habit had no significant impact on the consumption of various beverages by children. In turn, milk consumption decreased significantly in the study population. In 2018, children consumed 1.3 ± 1.1 servings of milk and 1.6 ± 1.2 servings of dairy beverages, including sweetened dairy beverages, on average. In 2023, the average milk consumption decreased to 0.9 ± 1.2 servings, whereas the consumption of sweetened and non-sweetened dairy beverages increased to 1.9 ± 1.5 servings. Therefore, the intake of added sugars from beverages increased in the analyzed period.

Fluids should be consumed regularly during the day, not only at mealtime. In the study population, 84% of girls and 78% of boys had access to beverages at school, but only 51% of girls and 44% of boys consumed beverages daily during the school day. In 2018, 25% of the surveyed students, on average, purchased beverages in school (sex was not a differentiating factor; *p* < 0.05), and the remaining respondents brought drinks from home, mostly juice, natural mineral water, and non-carbonated SSBs. In 2023, 37% of girls and boys bought beverages at school or on the way to school, mostly carbonated SSBs, flavored mineral water, and natural mineral water. According to 20% of the children, they were not prompted by their parents to take beverages to school, and more than 43% of the children were not encouraged to consume fluids at home. In 2018, most respondents were offered juice, sweetened hot beverages, and natural mineral water at home. In 2023, the children were offered mainly carbonated SSBs, sweetened hot beverages, and natural mineral water. 

The proportion of SSBs, in particular, flavored mineral water and juice in cartons, was significantly higher among overweight children. Body weight expressed as the BMI was not correlated with the consumption frequency of carbonated and non-carbonated SSBs, but children with a normal body weight consumed milk and dairy beverages significantly more often (1.31; 95% CI 1.03–1.56; *p* < 0.05). Children of less educated parents, younger parents, and parents residing in urban areas consumed more EDs. Average fluid intake expressed in milliliters was not significantly affected by BMI values, place of residence, parental education, or age. Gender was the only factor that exerted a significant effect on average fluid intake (Table 4 and Figure 2). Parental age and education were also significantly correlated with the children’s preference for carbonated SSBs.

Children’s awareness of the health impacts of various beverages was also analyzed in the study (Figure 3). Milk, dairy beverages, fruit and vegetable juice, tea, compote, and natural mineral water (including tap water) were recognized as healthy items. More than 2/3 of the children identified carbonated and non-carbonated SSBs as less healthy choices. Around 40% of boys and nearly 50% of girls classified carbonated mineral water as “less healthy”. At least a third of the children surveyed in 2018 and 25% of the children surveyed in 2023 regarded non-carbonated SSBs as “healthy”. The study demonstrated that children were aware of the negative health consequences of consuming SSBs. In 2023, 76% of boys and 83% of girls recognized that SSBs contribute to dental caries. The link between sugary drinks and overweight was recognized by 50% of boys and 65% of girls. No significant differences in the percentage of answers were noted between the analyzed years. 

## 4. Discussion

Children need to drink water throughout the day to maintain normal body temperature and produce bodily fluids that are essential for good health and daily functioning. Young children are at the highest risk of dehydration. According to Polish guidelines, girls should consume 1900 mL of fluids/day, and boys should consume 2100 mL of fluids/day [18]. In this study, only 1/4 and 1/5 of the children met their daily fluid intake requirements in 2018 and 2023, respectively. In 2018, boys consumed 210 mL more fluids than girls on average (1794 mL/day vs. 1583 mL/day; *p* < 0.05). Similarly, in 2023, boys consumed 250 mL more fluids than girls on average, but these values were lower than the recommended average fluid intake for this age group. In 2014, Fenández-Alvira et al. analyzed 238 Spanish children of both sexes and found that 87% of the respondents failed to meet daily water intake requirements [19]. The results of the US National Health and Nutrition Examination Survey (NHANES) conducted in 2005–2010 also demonstrated that more than 75% of 4- to 13-year-olds did not meet dietary water guidelines [20]. Kenney et al. analyzed NHANES data for 2009–2012 and reported that more than 50% of the participating children were inadequately hydrated [21]. A cross-sectional study of around 12,000 children and teenagers from 15 countries revealed that a large number of the participants in all age groups failed to meet EFSA recommendations for water intake [22]. 

A detailed analysis of fluid intake in six countries demonstrated that children consume only 14% of their total fluid intake at school, where they spend most of their day [23]. Similar observations were made in clinical studies that found that many children had highly concentrated urine before or during school [24,25], which suggests that many students arrive at school dehydrated and do not drink fluids during the school day. Bougatsas et al. reported that an individual’s hydration status is influenced not only by fluid intake but also by the type of consumed beverages. In children aged 8–14 years, a drinking pattern based on water and milk was associated with better hydration (lower urine osmolality), whereas regular consumption of carbonated drinks and other SSBs, but not water, was associated with inferior hydration [26]. 

Less is known about the relationship between obesity and hydration status in children. It should also be noted that water turnover rates increase with the BMI due to higher energy requirements, greater food consumption, and higher metabolic production [27,28]. Kozioł-Kozakowska et al. studied Polish children aged 7–15 years and found that 53% of the participants were insufficiently hydrated, whereas 16.3% were severely dehydrated during the school day (>1000 mOsm/kgH2O). Overweight children were more dehydrated. The risk of dehydration at school was 2.3 times higher in children with a high body fat percentage (BF%) [29]. Obese children were characterized by lower fluid consumption, lower total body water percentage, higher urine density, and lower hydration status than children with a healthy weight [30]. In the present study, body weight expressed as the BMI was not significantly correlated with total daily fluid intake. 

According to international dietary guidelines, school children should consume two to three servings of dairy products daily [31,32]. Despite dietary recommendations and the fact that cow’s milk is an important source of dietary nutrients, milk consumption among children in developed countries, such as Ireland, the USA, France, and Germany, has decreased in the past decade [31,32,33,34]. Similar observations were made in the current study, where milk consumption decreased, and the percentage of children meeting dietary guidelines for milk and dairy intake was reduced from 31% to 23% between 2018 and 2023. In a study conducted by the United Nations Food and Agriculture Organization (FAO), milk was the main school drink in only 28% of the surveyed countries. One of the reasons for the decline in milk intake is the promotional activity of competitive beverage brands. The FAO urged the dairy industry to “adopt an aggressive consumer-oriented marketing strategy to make milk and milk products attractive to school children” [35]. According to the US Departments of Agriculture and Health and Human Services, American children and adolescents older than four do not consume enough dairy to meet the recommendations in the federal Dietary Guidelines for Americans (DGA). Recently, a panel of experts from the Academy of Nutrition and Dietetics, the American Academy of Pediatric Dentistry, and the American Academy of Pediatrics concluded that milk—whole, low-fat, and skim—offers a host of essential nutrients that young children need to be healthy. The expert panel also recommended that parents strictly limit children’s intake of other beverages, except for water and small amounts of 100% juice [36].

The consumption of SSBs has risen significantly, which may lead to various health problems. The negative impact of SSB consumption on children and adolescents remains a significant public health problem. There is evidence to indicate that SSB intake is not only related to obesity, hypertension, and type 2 diabetes mellitus (T2DM) [37] but that it also affects the development of the nervous system in children [38,39]. According to Wesnes et al., the replacement of a typical breakfast with a sugary drink can lead to a decline in attention, concentration, and memory in children [40]. The consumption of SSBs was high in the study population, and the respondents’ intake of carbonated SSBs increased significantly between 2018 and 2023. Numerous research studies have examined the relationship between high sugar intake, in particular, added sugar from drinks, and a decline in concentration and attention in children, including children with ADHD. Lien et al. found that a high intake of SSBs was associated with hyperactivity among adolescents [41]. A study of American middle school students revealed that the risk of ADHD symptoms was 1.14 and 1.66 higher in children consuming SSBs and EDs, respectively [42]. Ptacek et al. examined eating patterns in Czech children and found higher levels of SSB consumption among boys diagnosed with ADHD [43]. However, the results of some studies are inconclusive. King and Chang analyzed the sugar intake of Korean primary school students, where only 8.5% of the participants were diagnosed with ADHD, and did not find a significant correlation between simple sugar intake from soft drinks and increased risk of ADHD [44].

The growing consumption of EDs among children and adolescents poses a serious problem. In the present study, ED intake increased between 2018 and 2023. A 2013 study of 37,500 children and adolescents from 16 European countries found that 18% of children (3–10 years) and 68% of adolescents (10–18 years) consumed EDs in the previous year [45]. Research has shown that 12–35% of children and young adults consume EDs at least once per week [46,47,48]. According to a report of the Polish National Institute of Public Health–National Institute of Hygiene, 2.1% of children aged 3–9 years consume EDs. The intake of EDs increases with age, and EDs are consumed by 27.4% of adolescent girls and by 35.7% of adolescent boys (10–17 years) [49]. The results of the present study were corroborated by Błaszczyk-Bębenek, who found that ED consumption among children and adolescents in the Polish voivodeship of Małopolska was influenced by sex. In the cited study, boys consumed significantly more EDs than girls and were more likely to buy EDs in larger packaging [50]. Caffeine, the main ingredient in EDs, is a psychoactive substance that should not be consumed by children. Numerous studies have demonstrated that high caffeine intake in children and adolescents can lead to sudden mood swings, irritation, and anxiety and can influence total sleep time and sleep quality [51,52,53]. Frequent ED consumption (more than seven times per week) was also positively associated with asthma, allergic rhinitis, and atopic dermatitis [54].

### Strengths and Limitations

This is the first study to examine changes in the fluid intake and beverage preferences of Polish adolescents between 2018 and 2023. The analyzed period covered the COVID-19 pandemic, which, according to research, led to changes in food availability and consumers’ eating preferences. The main strength of this study was that the parents’ and the children’s responses concerning beverage preferences and factors that affect these preferences were compared. This is a very important consideration in 11- to 13-year-olds who are beginning to make independent food choices and are susceptible to peer pressure and information posted on social media. The study revealed changes in the percentage of respondents who made healthy food choices, and it demonstrated that the parents’ choices often differ from the choices made by their children.

The main limitation of the study was that the questionnaire provided mostly qualitative information about fluid intake and beverage preferences in the study population. Quantitative data were used only to assess total fluid intake and the intake of water, SSBs, milk, and dairy beverages. Future research should involve food journals to evaluate the respondents’ intake of specific beverages and daily and weekly consumption frequency and to link the results with the participants’ sedentary behaviors, physical activity, and lifestyle. Another limitation of the study is that it does not include water intake from food, which, depending on the type of diet, can make up a significant proportion of the daily requirement. The study was concerned with assessing the intake of drinks consumed, which are not the only source of water in a child’s diet.

## 5. Conclusions

Daily fluid intake from beverages in children aged 11–13 years generally does not meet recommended intakes. Physically active children consumed more fluids, including natural mineral water, whereas sedentary children were more predisposed to consuming SSBs. Most children did not consume fluids regularly, and more than 25% of the respondents did not ask for a drink when they were thirsty. Low fluid intake can negatively affect children’s hydration status and bodily functions. Taste preferences and advertising were correlated with a higher intake of carbonated and non-carbonated SSBs and dairy beverages in both 2018 and 2023. The percentage of children who bought drinks independently and had access to SSBs, including EDs, increased significantly during the analyzed period (2018–2023). These results indicate that nutrition education programs are needed to teach adolescents to make healthy drink choices, limit their consumption of SSBs and EDs, and promote regular intake of natural mineral water and non-sweetened dairy beverages. 

## Figures and Tables

**Figure 1 nutrients-16-01625-f001:**
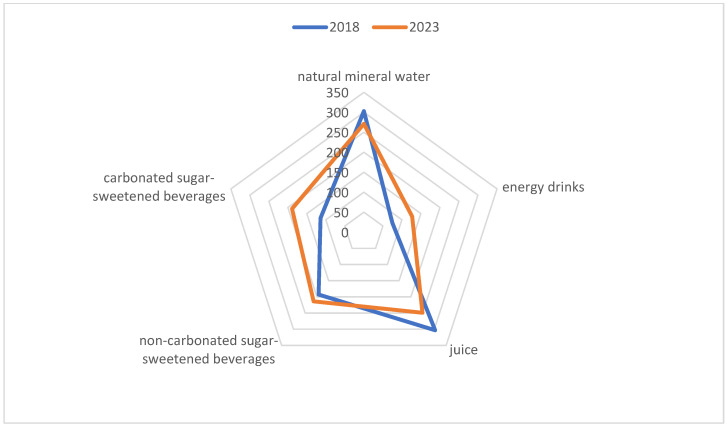
Changes in the number of children [*n*] consuming various beverages between 2018 and 2023.

**Figure 2 nutrients-16-01625-f002:**
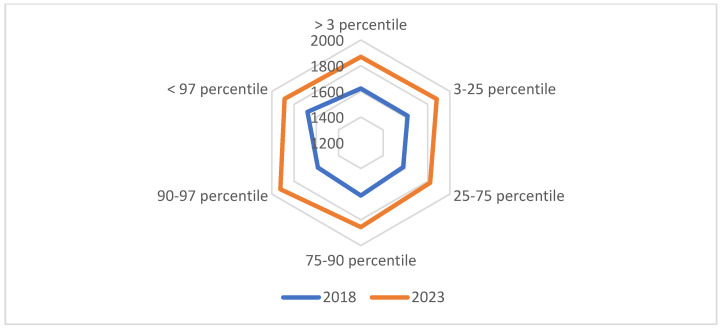
Average fluid intake expressed in mL/day relative to the BMI in 2018 and 2023.

**Figure 3 nutrients-16-01625-f003:**
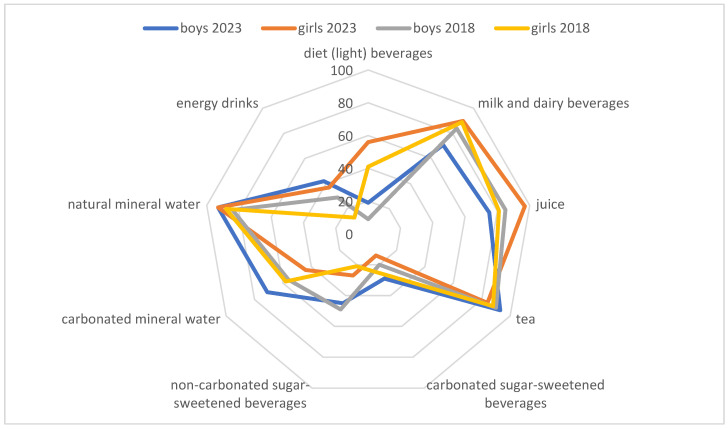
Awareness of the health impacts of various beverages among girls and boys in 2018–2023 (percentage of children who considered a given beverage to be healthy, %).

**Table 1 nutrients-16-01625-t001:** Demographic characteristics of the study population.

	2018 *n* = 491 [%]	2023 *n*= 539 [%]
Gender		
Boys	226 [46%]	243 [45%]
Girls	265 [54%]	296 [55%]
Age (years)		
11	156 [32%]	174 [32%]
12	171 [35%]	179 [33%]
13	164 [33%]	186 [35%]
Place of residence		
Urban area	317 [65%]	348 [65%]
Rural area	174 [35%]	191 [35%]
BMI		
Weight deficiency	112 [23%]	106 [20%]
Normal body weight	208 [42%]	314 [58%]
Excessive body weight	171 [35%]	119 [22%]

**Table 2 nutrients-16-01625-t002:** Fluid consumption and factors affecting beverage choices according to the parents and the children in 2018–2023.

	2018 *n* = 491			2023 *n* = 539			*p*-Value (Parents)	*p*-Value (Children)
	According to the Parents [%]	According to the Children [%]	*p*-Value	According to the Parents [%]	According to the Children [%]	*p*-Value		
Factors affecting beverage choices								
Taste preferences	363 [73.9]	411 [83.7]	<0.05	401 [74.4]	478 [88.7]	<0.05	ns	<0.05
Beverage composition	311 [63.3]	104 [21.2]	<0.01	384 [71.2]	131 [24.3]	<0.01	<0.05	ns
Sugar content	276 [56.2]	121 [24.6]	<0.01	301 [55.8]	154 [28.6]	<0.05	ns	ns
Price	297 [60.5]	78 [15.9]	<0.01	317 [58.8]	69 [12.8]	<0.01	ns	ns
Habit	137 [27.9]	354 [72.1]	<0.01	164 [30.4]	407 [75.5]	<0.01	ns	ns
Advertising	171 [34.8]	256 [52.1]	<0.05	196 [36.4]	315 [58.4]	<0.05	ns	<0.05
Health promotion	141 [28.7]	78 [15.9]	<0.05	159 [29.5]	109 [20.2]	<0.05	ns	<0.05
Beverage preferences								
Natural mineral water (include tap water)	286 [58.2]	303 [61.7]	ns	307 [56.9]	314 [58.3]	ns	ns	ns
Carbonated mineral water	79 [16.0]	117 [23.8]	ns	94 [17.4]	126 [23.3]	ns	ns	ns
Flavored mineral water	209 [42.6]	187 [38.1]	ns	176 [32.6]	123 [22.8]	<0.05	<0.05	<0.05
Non-carbonated sugar-sweetened beverages	207 [42.2]	193 [39.3]	ns	254 [47.1]	214 [39.7]	<0.05	<0.05	ns
Carbonated sugar-sweetened beverages	84 [17.1]	114 [23.2]	<0.05	113 [20.9]	189 [35.1]	<0.05	ns	<0.05
Energy drinks	11 [2.2]	34 [6.9]	ns	21 [3.9]	127 [23.5]	<0.05	ns	<0.05
Juice	256 [52.1]	303 [61.7]	<0.05	303 [56.2]	249 [46.2]	<0.05	ns	<0.05
Tea	211 [43.0]	156 [31.8]	<0.05	256 [47.5]	177 [32.8]	<0.05	ns	ns
Milk and dairy beverages	144 [29.3]	134 [27.3]	ns	134 [24.8]	159 [29.5]	ns	<0.05	ns
Compote	107 [21.8]	87 [17.7]	ns	121 [22.4]	103 [19.1]	ns	ns	ns
Diet (light) beverages	24 [4.9]	50 [10.2]	<0.05	76 [14.1]	101 [18.7]	<0.05	<0.05	<0.05
Daily fluid intake								
1–2 glasses	34 [6.9]	46 [9.4]	ns	41 [7.6]	49 [9.1]	ns	ns	ns
2–4 glasses	187 [38.1]	165 [33.6]	ns	207 [38.4]	183 [33.9]	ns	ns	ns
4–6 glasses	128 [26.0]	159 [32.4]	<0.05	181 [33.6]	209 [38.8]	ns	<0.05	<0.05
6–8 glasses	87 [17.7]	70 [14.3]	ns	79 [14.7]	67 [12.4]	ns	ns	ns
More than 8 glasses	53 [10.8]	51 [10.4]	ns	31 [5.7]	31 [5.7]	ns	<0.05	<0.05

ns—not statistically significant.

**Table 3 nutrients-16-01625-t003:** Odds ratios (95% confidence interval) in an analysis of the relationships between the consumption of various beverages by children and factors affecting the choice of beverages in the study populations in 2018–2023.

	Consumption of Flavored Mineral Water/Non-Carbonated Sugar-Sweetened Drinks (Ref. Consumption of Natural Mineral Water)		Consumption of Carbonated Sugar-Sweetened Drinks (Ref. Consumption of Natural Mineral Water)		Consumption of Energy Drinks (Ref. Consumption of Natural Mineral Water)		Consumption of Sugar-Sweetened Dairy Beverages (Ref. Milk Consumption)	
	OR (95% CI)	*p*	OR (95% CI)	*p*	OR (95% CI)	*p*	OR (95% CI)	*p*
2018								
Taste preferences	1.07 (0.94–1.12)	ns	1.31 (1.13–1.43)	<0.05	1.19 (1.03–1.27)	ns	1.54 (1.27–1.71)	<0.01
Beverage composition	1.04 (0.91–1.09)	ns	1.09 (0.94–1.17)	ns	1.08 (0.97–1.15)	ns	1.29 (1.04–1.37)	<0.05
Advertising	1.09 (0.91–1.21)	ns	1.29 (1.1–1.34)	<0.05	1.02 (0.89–1.16)	ns	1.34 (1.08–1.49)	<0.05
Health promotion	0.91 (0.84–1.03)	ns	0.91 (0.88–1.05)	ns	0.88 (0.74–0.96)	<0.05	1.44 (1.21–1.57)	<0.05
2023								
Taste preferences	0.78 (0.63–0.89)	<0.05	1.46 (1.17–1.62)	<0.05	1.39 (1.11–1.49)	<0.05	1.59 (1.34–1.71)	<0.01
Beverage composition	1.02 (0.91–1.17)	ns	1.07 (0.93–1.14)	ns	1.13 (1.02–1.25)	ns	1.45 (1.3–1.66)	<0.05
Advertising	1.34 (1.17–1.49)	<0.05	1.46 (1.23–1.59)	<0.05	1.12 (0.97–1.19)	ns	1.01 (0.93–1.15)	ns
Health promotion	0.74 (0.61–0.94)	<0.05	0.71 (0.53–0.88)	<0.05	0.74 (0.67–0.89)	<0.05	1.52 (1.29–1.63)	<0.01

ns—not statistically significant.

**Table 4 nutrients-16-01625-t004:** Odds ratios (95% confidence interval) in an analysis of the relationships of average fluid intake expressed in milliliters relative to the BMI between 2018 and 2023 between children’s beverage preferences in 2023 vs. demographic factors (sex, body mass) and parental factors (education, age, and place of residence).

	Boys (Ref. Girls)			Overweight (Ref. Normal Body Weight)			Parents with Secondary Education (Ref. Parents with University Education)			Parents Residing in Rural Areas (Ref. Parents Residing in Urban Areas)			Parents Aged ≤ 40 Years (Ref. Parents Aged > 40 Years)		
	OR	95% CI	*p*	OR	95% CI	*p*	OR	95% CI	*p*	OR	95% CI	*p*	OR	95% CI	*p*
Natural mineral water	1.44 *	1.01–1.69	0.03	0.86 *	0.76–1.02	0.041	0.78 *	0.71–0.92	0.026	0.77 *	0.61–0.89	0.023	0.98	0.89–1.07	ns
Carbonated mineral water	0.94	0.83–1.04	ns	1.03	0.91–1.09	ns	1.13	0.94–1.25	ns	1.03	0.90–1.12	ns	0.80 *	0.69–0.94	0.041
Flavored mineral water	1.13	0.97–1.21	ns	1.46 **	1.23–1.87	0.0072	1.11	1.03–1.21	ns	1.10	0.93–1.24	ns	1.11	1.02–1.15	ns
Non-carbonated sugar-sweetened beverages	1.27 *	1.02–1.44	0.023	1.29	1.02–1.39	0.031	1.34 *	1.11–1.46	0.028	1.12	0.99–1.21	ns	1.07	1.03–1.27	ns
Carbonated sugar-sweetened beverages	1.35 *	1.09–1.56	0.018	1.07	0.88–1.14	ns	1.39 *	1.09–1.56	0.026	0.94	0.81–1.03	ns	1.46 **	1.19–1.63	0.0081
Energy drinks	1.31 *	1.07–1.49	0.021	1.11	1.03–1.20	ns	1.31 *	1.12–1.48	0.028	0.71 *	0.63–0.89	0.019	1.27 *	1.09–1.44	0.039
Juice	0.74 *	0.58–0.86	0.016	1.78 **	1.34–2.11	0.0031	1.13	0.98–1.23	ns	1.03	0.91–1.07	ns	1.33 *	1.13–1.48	0.021
Tea and other sweetened hot beverages	1.33 *	1.12–1.46	0.025	1.08	0.93–1.14	ns	1.07	0.92–1.24	ns	1.07	1.01–1.16	ns	1.12	0.98–1.17	ns
Milk and dairy beverages	0.88 *	0.71–1.04	0.038	0.76 *	0.66–0.89	0.027	0.76 *	0.64–0.91	0.024	1.12	1.03–1.24	ns	0.71 *	0.63–0.89	0.028
Fluid consumption per day	1.29 *	1.11–1.48	0.031	1.04	0.87–1.27	ns	1.11	0.89–1.21	ns	1.09	0.91–1.26	ns	1.03	0.91–1.19	ns

ns—not statistically significant. * *p* < 0.05, ** *p* < 0.01 in Wald’s test.

## Data Availability

Due to ethical restrictions and participant confidentiality, data cannot be made publicly available.

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
