# Peer review of "Beverage Consumption and Factors Influencing the Choice of Beverages among Polish Children Aged 11–13 Years in 2018–2023"

_nutrients, 2024, doi:10.3390/nu16111625_

Round 1

Reviewer 1 Report

Comments and Suggestions for Authors

1.       Line 29: as it is the first mentioning of “SSB”, the full name “sugar-sweetened-beverages should be given. Later it is not necessary anymore

2.       Lines 63 – 69: These sentences belong to discussion and not in the introduction.

3.       Lines 102 – 107. The questionnaire could be given as supplementary information in case someone likes to read it.

4.       Lines 145 – 153. The fluid intake was presumable ml/ Day? If so it should be written

5.       Figure 1. The increasing display of the numbers are presumably ml? It should be explained on the figure or in the legend.

6.       Lines 180-182: Was the increased water intake in boys higher in sports activity or in general? Unclear!

7.       Table 2. Most data shown in table 2 are insignificant. Therefore, the table should be condensed to those values, which were significantly different. The insignificant data could just be mentioned in the text.

8.       Table 3. Is it really necessary to give these Data? Could be easily given in the text. It is a little tedious to read that table.

9.       One important aspect of the study has not been answered: did the children tested during two periods meet the national or international water intake requirements? Nothing has been mentioned in the results. It is mentioned in the discussion, but it should be part of the results, because it is an important finding.

10.   Line 294: This statement again should better be explained in the result section. Body mass means weight? Better would be body mass index, which could be easily calculated as age, weight and height of each subject was known. Recalculation of fluid consumption per day according to body mass index is essential. The result could well be displayed by a graph, rather than a table.

11.   The classical Polish family meal contained usually the consumption of a soup during the meal. Is it still the case? And if so have these children got some additional fluid intake by a soup per day?

Author Response

Thank you for reviewing our article. Your feedback and insights were extremely valuable in improving the quality of the manuscript. We appreciate the time and effort you took to evaluate our work. Below are our detailed responses the Reviewer’s comments. All changes made in the manuscript are marked in green.

Line 29: as it is the first mentioning of “SSB”, the full name “sugar-sweetened-beverages should be given. Later it is not necessary anymore

The full name was provided.

 Lines 63 – 69: These sentences belong to discussion and not in the introduction

Thank you for this suggestion, the paragraph was reworded and moved to the Discussion section.

Lines 102 – 107. The questionnaire could be given as supplementary information in case someone likes to read it.

The survey was described in detail, further details were added in the Materials and Methods section

Lines 145 – 153. The fluid intake was presumable ml/ Day? If so it should be written

The missing information was provided: /day

Figure 1. The increasing display of the numbers are presumably ml? It should be explained on the figure or in the legend.

Fig. 1. The increasing numbers represent the number of respondents consuming various beverages. It was explained in the legend: [n]

Lines 180-182: Was the increased water intake in boys higher in sports activity or in general? Unclear!

The relevant explanation was provided.

Table 2. Most data shown in table 2 are insignificant. Therefore, the table should be condensed to those values, which were significantly different. The insignificant data could just be mentioned in the text.

Thank you for this suggestion, Table 2 was condensed and the factors that had no significant impact on the choices and consumption of beverages were mentioned in the text.

Table 3. Is it really necessary to give these Data? Could be easily given in the text. It is a little tedious to read that table.

Thank you for this comment. However, please note that it is not easy to describe large amounts of data in the text, which is why we decided to present some results in tabular form. We have used this form of presentation in our previous articles to maintain a balance between the data described in the text and presented in tables.

One important aspect of the study has not been answered: did the children tested during two periods meet the national or international water intake requirements? Nothing has been mentioned in the results. It is mentioned in the discussion, but it should be part of the results, because it is an important finding.

Thank you for this suggestion. This sentence was moved to the Results section where we provided data on fluid intake.

Line 294: This statement again should better be explained in the result section. Body mass means weight? Better would be body mass index, which could be easily calculated as age, weight and height of each subject was known. Recalculation of fluid consumption per day according to body mass index is essential. The result could well be displayed by a graph, rather than a table.

Thank you for this suggestion. The missing data were provided in Table 3, which presents the results for overweight children compared to normal body weight children. Figure 2 was also added.

The classical Polish family meal contained usually the consumption of a soup during the meal. Is it still the case? And if so have these children got some additional fluid intake by a soup per day?

Currently, the consumption of soups is not common in the child population. We did not analyze the consumption of soups because the aim of our study was to assess changes in fluid intake, with particular emphasis on still water and sweetened drinks. Nevertheless, we agree that soup is a source of fluids.

Reviewer 2 Report

Comments and Suggestions for Authors

Abstract: Remove the word between 2018 to 2023 from line19.

Conclusion is missing. Add conclusion in the abstract.

Introduction:

I feel that the introduction section need  to be revised. It is written like a discussion section. I did not find coherence and connection between different sentences. I am happy to accept a small introduction, but it should only gives a background with a solid rationale.

Aim of study:

Remove this heading "Aim of study"

Remove between 2018 to 2013 from line 80.

The rationale is weak.

Methods:

From 97, create a new subheading, Study questionnaire or data collection method. I would highly encourage if you add subheadings, rather than presenting everything under one subheading.

Content of subheading data collection is more or less related to the content presented in line 97 to onwards. 

Follow STROBES guidelines and write subheading in the method section accordingly.

I did not see study setting, study design ( i am confused either it was a cross sectional or cohort), sampling methods. Add these subheadings in your method section.

Which model did you use? It is not clear which test you used. Bivariate or multivariate. I think you applied logistic regression because the study measure is non-continous.

Results:

Add a study demographic table.

In the results, I found that you also have written a few sentences which aren't part of result. Write only what you have in figure and table. Rest of the things, you can discuss in discussion.

Line 247-248, should be part of methods.

What do you mean by most children, write in % in line 248.

Remove the line 249-250, if it is part of result write its significace (p-value or odds).

Result of the things seems appropriate.

Discussion:

Start discussion with your study findings rather than citing others, then discuss others authors findings.

Author Response

Thank you for reviewing our article. Your feedback and insights were extremely valuable in improving the quality of the manuscript. We appreciate the time and effort you took to evaluate our work. Below are our detailed responses the Reviewer’s comments. All changes made in the manuscript are marked in blue.

Abstract: Remove the word between 2018 to 2023 from line19.

Conclusion is missing. Add conclusion in the abstract.

Thank you for these comments, the relevant changes were made.

I feel that the introduction section need  to be revised. It is written like a discussion section. I did not find coherence and connection between different sentences. I am happy to accept a small introduction, but it should only gives a background with a solid rationale.

Thank you for this comment. The Introduction section was shortened and rewritten. We hope that in its current form, it better corresponds to the topic and meets the Reviewer’s and readers’ expectations.

Remove this heading "Aim of study"

Remove between 2018 to 2013 from line 80.

The rationale is weak.

The objective of the study was described in greater detail.

From 97, create a new subheading, Study questionnaire or data collection method. I would highly encourage if you add subheadings, rather than presenting everything under one subheading.

Content of subheading data collection is more or less related to the content presented in line 97 to onwards. 

Follow STROBES guidelines and write subheading in the method section accordingly.

I did not see study setting, study design ( i am confused either it was a cross sectional or cohort), sampling methods. Add these subheadings in your method section.

Thank you for this suggestion. To increase the clarity of presentation in the Materials and Methods section, subsections were added and more details regarding the study were provided.

Add a study demographic table.

In the results, I found that you also have written a few sentences which aren't part of result. Write only what you have in figure and table. Rest of the things, you can discuss in discussion.

We added table.

Some of the results are described in the text to complement the analyses presented in the form of tables and graphs. We have used this form of presentation in our previous articles as it makes it easier for the readers to follow and interpret the results, including those who are not involved in scientific research on a daily basis.

Line 247-248, should be part of methods.

What do you mean by most children, write in % in line 248.

Remove the line 249-250, if it is part of result write its significace (p-value or odds).

Thank you for these suggestions, the relevant changes were made.

Start discussion with your study findings rather than citing others, then discuss others authors findings.

Thank you for this suggestion, the first paragraph of the Discussion section was reworded.

Round 2

Reviewer 1 Report

Comments and Suggestions for Authors

I have only two little changes which should be made:

1.  In the legend of Figure1 is still missing: ml/day

2. The same in Figure 2: ml/day missing

Author Response

Thank you for reviewing our article. Your feedback and insights were extremely valuable in improving the quality of the manuscript. We appreciate the time and effort you took to evaluate our work.

  1. I have only two little changes which should be made: Figure 1. Changes in the number of children [n] consuming various beverages between 2018 and 2023.

The results shown in Figure 1 and Figure 2 refer to the number of children and not the volume of fluid. In the case of Fig 1, the figure shows the number of children [n] who consumed different types of beverages in 2018 and 2023. The figure caption and commentary sentence have been reworded to be clear and the changes have been highlighted in orange.

In the studied period, the number of children consuming natural mineral water and juice decreased by 11.5% and 23.3%, respectively (p<0.05), whereas a significant increase was noted in the number of children consuming carbonated SSBs (41.4%, p<0.05) and EDs (32%, p<0.05) (Table 2, Fig. 1).

Figure 1. Changes in the number of children [n] consuming various beverages between 2018 and 2023.

  1. The same in Figure 2: ml/day missing - indeed in the caption of figure 2 only millilitres were given, we have completed the description with ml/day

Reviewer 2 Report

Comments and Suggestions for Authors

Abstract: Adding one line doesn't mean conclusion. you need to add complete 1-2 sentences.

Author Response

Thank you for reviewing our article. Your feedback and insights were extremely valuable in improving the quality of the manuscript. We appreciate the time and effort you took to evaluate our work.

Adding one line doesn't mean conclusion. you need to add complete 1-2 sentences.

Thank you for this suggestion, we have added another summary sentence and reworded the content of Conclusion in Abstract (changes highlighted in maroon), which appears comprehensive and exhaustive.
